# Subtype-Specific Tumour Immune Microenvironment in Risk of Recurrence of Ductal Carcinoma In Situ: Prognostic Value of HER2

**DOI:** 10.3390/biomedicines10051061

**Published:** 2022-05-03

**Authors:** Julia Solek, Jedrzej Chrzanowski, Adrianna Cieslak, Aleksandra Zielinska, Dominika Piasecka, Marcin Braun, Rafal Sadej, Hanna M. Romanska

**Affiliations:** 1Department of Pathology, Chair of Oncology, Medical University of Lodz, 92-213 Lodz, Poland; julia.solek@umed.lodz.pl (J.S.); aleksandra.zielinska3@stud.umed.lodz.pl (A.Z.); marcin.braun@umed.lodz.pl (M.B.); 2Department of Biostatistics and Translational Medicine, Medical University of Lodz, 90-419 Lodz, Poland; jedrzej.chrzanowski@stud.umed.lodz.pl (J.C.); adrianna.cieslak@stud.umed.lodz.pl (A.C.); 3Department of Molecular Enzymology and Oncology, Intercollegiate Faculty of Biotechnology, Medical University of Gdansk, 80-210 Gdansk, Poland; dominika.piasecka@gumed.edu.pl

**Keywords:** DCIS, recurrence, tumour immune microenvironment, HER2

## Abstract

Increasing evidence suggests that the significance of the tumour immune microenvironment (TIME) for disease prognostication in invasive breast carcinoma is subtype-specific but equivalent studies in ductal carcinoma in situ (DCIS) are limited. The purpose of this paper is to review the existing data on immune cell composition in DCIS in relation to the clinicopathological features and molecular subtype of the lesion. We discuss the value of infiltration by various types of immune cells and the PD-1/PD-L1 axis as potential markers of the risk of recurrence. Analysis of the literature available in PubMed and Medline databases overwhelmingly supports an association between densities of infiltrating immune cells, traits of immune exhaustion, the foci of microinvasion, and overexpression of HER2. Moreover, in several studies, the density of immune infiltration was found to be predictive of local recurrence as either in situ or invasive cancer in HER2-positive or ER-negative DCIS. In light of the recently reported first randomized DCIS trial, relating recurrence risk with overexpression of HER2, we also include a closing paragraph compiling the latest mechanistic data on a functional link between HER2 and the density/composition of TIME in relation to its potential value in the prognostication of the risk of recurrence.

## 1. Introduction

Ductal Carcinoma in Situ (DCIS), the non-obligate precursor of invasive ductal breast carcinoma (IDC), is the most frequent type of non-invasive breast cancer (BC), accounting for approximately 20–25% of the reported new BC cases [1]. Similar to its invasive counterpart, DCIS is not one entity but represents a heterogeneous group of pre-invasive breast lesions with different phenotypes and a variable propensity for recurrence [1]. Breast-conserving surgery (BCS) with or without radio and hormonal therapy remains to date the standard of care for newly diagnosed patients. It is estimated that approximately 16–22% of DCIS treated with BCS alone will relapse [2]. Half of the recurrences will occur as invasive disease, with ipsilateral tumours at least twice as common as contralateral lesions [3]. Postoperative radio or systemic therapy considerably reduce this rate (to 7–9%) but the management of the patients remains difficult due to lack of reliable prognostic and predictive biomarkers [4]. Routine assessment of clinicopathological parameters, such as nuclear grade, growth pattern, or surgical margins, deemed to predict recurrence after BCS, is inadequate and many patients are still under or overtreated [5]. Several factors have been proposed as candidate predictors of the risk of disease recurrence. These include collagen 11A1, cathepsin A, the thioredoxin-interacting protein (TXNIP), an integrin Lvβ6 expressed by myoepithelial cells, and markers of proliferation (p53, Ki-67, and cyclin D), whose prognostic value in DCIS has been demonstrated by a number of single analyses carried out in various cohorts [4,6,7,8,9,10,11,12,13]. However, from the wealth of available reports on putative indicators of disease recurrence, it is Human Epidermal Growth Factor Receptor 2 (HER2) that emerges as the most promising prognosticator [14,15,16,17,18]. Results of the largest biomarker study, namely the UK/ANZ DCIS randomized trial, demonstrated that increased recurrence risk, particularly the risk of in situ ipsilateral second breast event (SBE), was associated with overexpression of HER2 [19]. Furthermore, overexpression of HER2 and ER proved to be predictive of radiotherapy and the tamoxifen benefit, respectively [20]. However, the biological role of the potential prognostic biomarkers investigated to date remains unclear and none of the above has been incorporated into clinical practice. There is a pressing need for identification of DCIS traits that would reliably enable prognostication and stratification of the patients for adequate therapeutic regimens.

The importance of bidirectional interactions between a tumour and its microenvironment as key modulators of disease evolution is, today, widely recognized. Immune cells, a consistent part of TME, are of particular clinical interest in various cancers, including BC, as a potential prognostic adjunct and a predictor of response to targeted therapies. Stimuli derived from the tumour immune microenvironment (TIME), in particular those induced by tumour-infiltrating lymphocytes (TILs) and tumour-associated macrophages (TAMs), are thought to be of critical significance for tumour development. Different frequencies of immune cells have been identified in each BC molecular subtype, suggesting that specific TIME might be associated with a distinct patient outcome [21,22,23,24]. As immune infiltration is most prominent at the early stages of BC evolution [24], it has been suggested that TIME may orchestrate “immune escape” (a switch to a less active tumour immune environment) and, hence, considerably contribute to the DCIS→IDC progression, which is considered a key step of immunoediting in breast cancer [25,26]. However, despite extensive research efforts, as summarized in a series of recent reviews, a role of immune cells in the DCIS course that would define subtype-specific tumour behaviour and patients’ prognosis is still unclear [27,28,29,30,31,32,33,34,35,36,37,38,39,40,41]. Here, an updated review of existing data on a role of the HER2–TIME crosstalk in DCIS and its potential value in the stratification of the risk of recurrence is presented in the context of the available evidence of the subtype-dependent clinical significance of stromal immune cells in DCIS (summarized in Table 1).

## 2. Molecular Subtypes of Breast Carcinoma: Immune Characterization

Human breast cancer represents a spectrum of tumour subtypes with distinct morphology, natural history, and responsiveness to therapy. Based on a seminal study by Perou et al., four main molecular subtypes of BC, related to the expression of the estrogen receptor (ER), the progesterone receptor (PR), and HER2 (ER/PR/HER2 status), have been distinguished: Luminal A, Luminal B, HER2-positive, and Triple Negative (TN) [42]. All these subtypes are present in both DCIS and IDC albeit with different frequencies. In both DCIS and IDC, HER2-positive and TN subtypes are frequently associated with a high histopathological grade and several genetic events such as p53 dysfunction, aneuploidy, Ras protein overexpression, and poor prognosis, while luminal subtypes are usually found to be of low grade [24,43].

Recent studies showed that in DCIS, the composition of the TIME depends on the molecular subtype. High levels of TILs are associated with ER-negativity and HER2-positivity, as well as with the basal phenotype and histological high grade [22,28,29,31,34,39,44,45,46,47,48]. Higher densities of Regulatory T cells (Tregs), activated both macrophages (CD68+PCNA+) and T cells (HLA-DR+), CD4+ T cells, CD20+ B cells, and total number of TILs are found in high-grade rather than in low-grade DCISs [49].

The possible molecular basis for a link between the composition/density of TIME and BC subtype has been disclosed by studies in invasive BC. In particular, in HER2-positive IDC, Gil Del Alcazar et al. demonstrated a negative association between the co-amplification of the 17q12 chemokine cluster with ERBB2 and the infiltration by activated T (GZMB+CD8+) cells [22]. When intrinsic subtyping using the PAM50 signature was applied, the co-amplification was found to be more common in PAM50 luminal HER2-positive IDC than in PAM50 HER2-enriched tumours. This link between HER2-positivity and migration/infiltration by TILs, suggestive of the co-evolution of BC cells and leukocytes during BC development, may also have clinical implications. As the presence of activated CD8+ TILs is associated with better response to both chemo and HER2-targeted therapies, their lower frequency in luminal IDCs could explain the probability of treatment resistance [22]. However, due to the functional diversity of chemokines in this cluster, the precise mechanism of their impact on TILs activity remains to be revealed. On the other hand, as demonstrated by several studies, including our recent work, HER2-postivity determines the BC cell response to inflammatory cytokines [23,50,51]. In various experimental settings, IL-6, TNF-α, and IL-1β were found to induce proliferative and secretory BC activities as well as affect stem-like properties that differed between HER2-negative and HER2-positive BC clones [23,50].

## 3. Pathological Evaluation of TILs in DCIS: Approaches and Current Clinical Utility

The immune contexture of the tumour (i.e., the location, density, and functional orientation of different subsets of immune cells) and its impact on clinical outcome are today at the forefront of translational research in various cancers [52,53,54]. For example, in colorectal cancer, studies by Galon et al. have indicated that the evaluation of intra- and peri-tumoral immune infiltration provides information more valuable than the widely accepted and routinely used AJCC/UICC-TNM classification [55].

In order to establish a standardized methodology for the evaluation of TILs for both clinical and research settings, in 2014, an International TILs Working Group composed of pathologists, medical oncologists, immunologists, and statisticians was created [56,57]. Quantitative assessment is routinely performed in hematoxylin and eosin (H&E) BC sections. Although initially set for IDC, the main recommendations, i.e., the inclusion of only stromal TILs (TILs’ density defined as the percentage of the stroma that is occupied by TILs), exclusion of tumour zones with crush artefacts or necrosis, and assessment of TILs as a continuous parameter, have also proved to be useful to the analysis of DCIS [23,27,32,40,46]. Among several other scoring methods applied to the evaluation of H&E-stained DCIS sections, quantification of ‘touching lymphocytes’, defined as lymphocytes that touch the basement membrane or are located from it within one lymphocyte cell’s thickness, has been shown by Toss et al. to be an independent prognostic marker [34]. Qualitative evaluation of immune cells’ infiltration involves analysis of tumour sections stained for a panel of specific markers that in DCIS most typically includes those described in the paragraph below. A summary of the following paragraphs is contained in Table 1.

**Table 1 biomedicines-10-01061-t001:** Reported associations between components of TIME, clinicopathological variables, and risk of recurrence. TIME—tumour immune microenvironment; ER—estrogen receptor; HER2—Human Epidermal Growth Factor Receptor 2; TILs—tumour-infiltrating lymphocytes; PD-1/PD-L1—Program Death Receptor/Ligand; TILB—tumour-infiltrating lymphocytes B; N—cohort size and number of reference in brackets.

TIME: Related Risk of Recurrence	Clinico-Pathological Variables	High TILs	High CD8	High CD4	High FoxP3	High PD-1/PD-L1	High TILB	High Macrophages
increased	HER2 +	Pruneri [27]; N = 1488 (2016), Thike [37]; N = 198 (2019)	Campbell [49]; N = 117 (2017), Thike [37]; N = 198 (2019), Semeraro [35]; N = 248 (2016)	Thike [37]; N = 198 (2019)	Toss [45]; N = 700 (2020)	Toss [45]; N = 700 (2020)	Milgy [36]; N = 80 (2017)	
	ER -	Darvihian [28]; N = 688; (2019), Toss [34]; N = 816 (2017)						Chen [46]; N = 80 (2020)
Not reported	High grade	Darvishian [28]; N = 688; (2019), Pruneri [27]; N = 1488 (2016), Toss [34]; N = 816 (2017), Morita [38]; N = 46 (2017), Beguinot [40]; N = 129 (2018), Thike [37]; N = 198 (2019) Chen [41]; N = 198 (2021), Toss [45]; N = 700 (2020) Semeraro [35]; N = 248 (2016) Hendry [32]; N = 138 (2017)	Campbell [49]; N = 117 (2017), Beguinot [40]; N = 129 (2018), Semeraro [35]; N = 248 (2016)	Campbell [49]; N = 117 (2017), Beguinot [40]; N = 129 (2018), Chen [41]; N = 198 (2021)	Campbell [49]; N = 117 (2017), Chen [41]; N = 198 (2021), Toss [45]; N = 700 (2020)	Chen [41]; N = 198 (2021), Toss [45]; N = 700 (2020) Thompson [29]; N = 27 (2016), Hendry [32]; N = 138 (2017)	Campbell [49]; N = 117 (2017), Beguinot [40]; N = 129 (2018)	Capmbell [49]; N = 117 (2017), Chen [46]; N = 80 (2020), Chen [41]; N = 198 (2021)
	Increased tumour size	Campbell [49]; N = 117 (2017), Darvishian [28]; N = 688; (2019)		Campbell [49]; N = 117 (2017)			Milgy [36]; N = 80 (2017) Campbell [49]; N = 117 (2017)	
	Presence of microinvasion	Morita [38]; N = 46 (2017), Toss [34]; N = 816 (2017)	Lv [39]; N = 85 (2019), Alcazar [22]; N = 36 (2017), Beguinot [40]; N = 129 (2018)	Beguinot [40]; N = 129 (2018)	Chen [41]; N = 198 (2021), Beguinot [40]; N = 129 (2018)	Lv [39]; N = 85 (2019), Beguinot [40]; N = 129 (2018)	Milgy [36]; N = 80 (2017)	Chen [46]; N = 80 (2020), Beguinot [40]; N = 129 (2018)
	High mitotic index	Campbell [49]; N = 117 (2017)	Beguinot [40]; N = 129 (2018)	Beguinot [40]; N = 129 (2018), Campbell [49]; N = 117 (2017)	Beguinot [40]; N = 129 (2018)	Beguinot [40]; N = 129 (2018)	Campbell [49]; N = 117 (2017)	Beguinot [40]; N = 129 (2018) Campbell [49]; N = 117 (2017)

## 4. Standard Parameters of TIME

### 4.1. CD8+ T Cytotoxic Cells

CD8 is a glycoprotein present on cytotoxic T lymphocytes. In IDC, higher numbers of CD8+ TILs are shown by different studies to be associated with diverse clinicopathological features and most frequently are predictive of favourable clinical outcome [58,59].

In DCIS, the decreasing signature of CD8+ TILs seems to signify increasing BC invasiveness and, hence, the process of the DCIS-to-IDC transition [22,60]. However, it has been now widely acknowledged that the impact of CD8+ TILs on disease prognosis depends not only on their density but also on the interplay with other TIME components as well as on the state of activation/exhaustion. Using recursive partitioning and regression tree analysis, the authors further showed that the decreased density of activated CD8+HLA-DR+ TILs or increased density of non-activated cytotoxic CD8+HLA-DR− TILs combined with high numbers of CD115+ macrophages was predictive of the risk of recurrence [49]. Therefore, as demonstrated below, depending on the state of activation of CD8+ TILs, different associations with prognostic variables have been noted.

A study demonstrated by Morita et al. indicated that the increasing number of CD8+ TILs might play a favourable role by promoting ‘healing’ of the lesions [38]. The authors demonstrated a link between high CD8+ TILs, high grade, comedo necrosis, apocrine features and signs of lesion regression in HER2-positive and TN DCIS subtypes, suggesting that CD8+ TILs may trigger the phenomenon of spontaneous healing and that the process is subtype-specific.

On the other hand, several studies revealed an opposite role of CD8+ TILs, which is reflected by their association with poor clinicopathological variables. Analysis of various TILs subsets in DCIS with and without microinvasion showed that CD8+PD1+ TILs were more frequent in DCIS with microinvasion regardless of the ER status, whereas CD8+PD1- TILs were the only subset whose density differed between HER2-positive and HER2-negative patients [39]. Campbell et al. demonstrated that CD8+ TILs were correlated with high grade, high Ki67, ER-negativity, and HER2-postitivity [49]. Gil Del Alcazar et al. showed a significant decrease in the number of activated GZMB+CD8+ TILs in IDC compared to DCIS, indicating the importance of CD8+ TILs functions in the process of DCIS progression. Interestingly, the authors also showed that the TCR clonotype was significantly lower in IDC than in DCIS, indicating that the loss of CD8+ TILs clonality might importantly participate in the process of invasion [22].

CD8+ TILs have also been implicated in the course and prognosis of HER2-positive subtypes. A study by Datta et al. revealed that high expression of HER2 was associated with increased expression of Interferon-γ/Tumour Necrosis Factor-α (IFN-γ/TNF-α) receptors, suggesting that the HER2-positive subtype might be particularly susceptible to T cell cytokine-mediated apoptosis [61]. An association between increased infiltration of CD8+ cells, the HER2-positive subtype, and increased risk of recurrence was also reported by several recent studies [35,37,49].

Taken together, available evidence seems to suggest that CD8+ TILs play an important role, particularly in HER2-positive DCIS. However, it should be emphasized that CD8+ TILs represent a heterogenous population and little is known about the clinical significance of their functional status or spatial location. For example, CD8+ TILs positive for CD103, the so called tissue-resident memory T cells (TRMs), have been recently identified as key immune players in TME. Their increased densities in the cancer island within IDC were significantly associated with relapse-free survival (RFS) but their prognostic value in DCIS is still yet to be revealed [62].

### 4.2. CD4+ T Helper Cells (CD4+ Th Cells)

CD4+ T helper cells include a variety of subpopulations which can stimulate immunoprotective or immunosuppressive reactions. For example, the immunosuppressive subpopulation of Th2 cells, through the production of the cytokine IL-10, can stimulate B cell activation or the polarization of macrophages towards the M2 (protumourigenic) phenotype. In contrast, the subpopulation of Th1 cells produce immunoprotective cytokines such as IFN-γ which stimulate the activation of cytotoxic CD8+ cells and the polarization of macrophages towards the M1 (antitmourigenic) phenotype [63]. The CD4+ TILs subsets display context-dependent phenotypic and functional plasticity that is influenced by oncogenic drivers and the presence of inflammation. Given appropriate environmental signals, the loss of the homeostatic balance between the subsets may lead to both abnormal auto- and anti-tumoral immune responses [64]. CD4+ cells were the only immune cells, the distribution of which did not differ between pure DCIS and DCIS with synchronous IDC. This might indicate that CD4+ cells are pertinent particularly for the early stages of DCIS evolution [45]. Infiltration with CD4+ cells in DCIS has been shown to be positively associated with HER2 positivity, ER negativity, high grade, and, in some studies, with a microinvasion [37,40,41,49,60]. While their prognostic significance in DCIS has not been unequivocally established [40,45,49], Thike et al. revealed that high densities of CD4+ TILs efficiently predict shorter disease-free survival [37]. Moreover, they indicated that the prognostic value of CD4+ cells might be influenced by other immune cells and showed that, if the balance between T helper and T cytotoxic cells is skewed towards the former (defined as a high CD4+/CD8+ ratio), the risk of recurrence is increased [37]. Accordingly, Datta et al. showed that in the HER2-positive subtype, the process of progression was associated with a progressive loss of T helpers cells’ function and a decreased production of IFN-γ [61]. Thus, it appears that CD4+ TILs, ‘the primary immune envoys’ to the stroma of DCIS lesions, particularly those overexpressing HER2, induce the immunosuppressive microenvironment, which is likely to be critical for further disease evolution.

### 4.3. CD4+CD25+FOXP3+ Regulatory T Cells (T Reg Cells)

Regulatory T cells (T reg cells) are derived from a lineage shared with naive CD4+ T cells and are characterized by expression of CD4, CD25, and FOXP3 [65]. Known as suppressor T cells, Tregs are thought to mediate negative immune responses [66]. Studies in IDC showed that dense FOXP3+ Tregs’ infiltration was associated with both poor recurrence-free survival and the HER2-positive subtype [67]. Similarly, recent reports indicate that in DCIS, a high density of immunosuppressive FOXP3+TIL cells was associated with the HER2-positive subtype, microinvasion, and high grade, and, consequently, was prognostic of an increased risk of recurrence [25,35,45]. Treg cells have been implicated in the process of immune evasion and the suppression of CD8+ T cell cytotoxic function, and, hence, affect the prognostic value of both CD4+ and CD8+ TILs [65]. Accordingly, in DCIS, the high FOXP3+/CD8+ ratio significantly correlated with an increased risk of recurrence [35]. The FOXP3+/CD8+ ratio, partially reflecting the balance between immunoprotection and immunosuppression, seems to be inherently linked to the molecular subtype. As shown by Thompson et al., a disturbed balance between CD8+ and FOXP3+ cells was found in ER-negative DCIS when compared to ER-positive lesions [29]. A study by Abba et al. showed that more aggressive DCIS (high grade and TN or HER2-positive) displayed signatures characteristic of activated Treg cells (CD4+/CD25+/FOXP3+), consistent with an immunosuppressive phenotype [30].

### 4.4. Markers of Immune Exhaustion of T Cells

Program Death Receptor 1 and Program Death Ligand 1 (PD-1 and PD-L1), which are known as immune checkpoints, regulate responses of the immune system and suppress T cell activity [68]. Upregulated expression of PD-1 or PD-L1 on T cells and/or tumour cells, reflecting ‘T cell exhaustion’ [69], is recognized in cancer as one of the major mechanisms of immune escape. In IDC, the expression of PD-L1 and PD-1 in both stromal and tumour compartments was shown to be associated with the HER2-positive subtype and ERBB2 amplification [70]. Tumours overexpressing the Epithelial Growth Factor Receptor (EGFR)/HER2 are characterized by elevated PD-L1 expression, suggesting that immune escape might be, at least in part, regulated by EGFR/HER2 signalling [70,71,72]. Moreover, PD-L1 expression could be decreased in a time/dose-dependent manner by treatment with HER2-targeting agents (lapatinib) [73], which supports the postulated dynamic crosstalk between TIME and tumour cells, especially those of the HER2-positive phenotype. In DCIS, several studies confirmed a similar phenotypic dependency of immune checkpoints and showed that the expression of PD-L1 and PD-1 on immune cells was significantly higher in the HER2-positive subtype [31,32,34,39]. Accordingly, CD8+ cells with high expression of inhibitory receptors (PD-1, CTLA-4, CD160, and CD244) were found to be associated with high-grade lesions [74]. Immune escape was also implicated in the DCIS-to-IDC transition as tumour cells in the areas of microinvasion were shown to upregulate the expression of PD-L1 [22,39]. Although available evidence strongly implicates the PD-1/PD-L1 axis in the progression of both IDC and DCIS, its prognostic significance still remains equivocal [75,76]. This is most likely due to the heterogeneity of the TIL population in terms of distinct states and profiles of exhaustion, the definition of which remains a major challenge. Results of recent studies using single-cell RNA sequencing confirm a remarkable diversity of immune cells, suggesting that distinct TIL states may have different impacts on disease course and outcome [77].

### 4.5. Tumour Infiltrating B Cells (TILBs)

Humoral immune responses against cancer are supported by several studies indicating that TILBs may play a role in the course and progression of DCIS. All mature B cells are typically identified by the presence of CD20 [78]. Other markers, i.e., CD38, CD138, and CD19, are expressed on plasma cells and cells regulating B cell development [78]. As demonstrated above for T cells, CD20+ TILBs were shown to be associated with HER2-positivity [45,49]. Interestingly, a study by Miligy et al. indicated that B lymphocytes may be inversely correlated with the process of invasion, showing that pure DCIS had a higher count of TILBs compared to the cases with invasive components [36]. Results of the study by Beguinot et al. further confirmed that the TILs/TILBs ratio was the only significant difference between DCIS without and with microinvasion, and was found to be significantly higher in the latter [40]. In contrast to DCIS, higher densities of TILBs detected in IDC have been associated with better survival, especially in ER-negative, TNBC, and HER2-positive tumours. However, the role of TILBs in breast cancer is largely understudied and, hence, their clinical value as a prognostic marker still remains unclear [36,45,49]

### 4.6. Macrophages

Tumour-associated macrophages (TAMs), derived from mononuclear cells, are the most abundant population of tumour-infiltrating cells in TME [79]. Tissue-resident macrophages are key regulators during mammary gland development, suggesting that normal mammary epithelial cells cooperate with these innate immune cells [80,81]. TAMs are recruited from the bloodstream in response to the chemotactic signals produced by both cancer cells and cellular components of TME [81]. Macrophages display high phenotypic and functional plasticity. They have two polarization states, namely classically activated M1 and alternatively activated M2 subtypes that, for TAMs, have been associated with anti- and pro-tumourigenic activities, respectively. However, this classification has been challenged by demonstration that, in fact, the TAMs population is very plastic and the two M1 and M2 states represent the extremes in a polarization continuum highly influenced by a plethora of microenvironmental factors [82,83,84]. TAMs are typically identified by positivity for CD68, CD163, and CD115 [82]. In DCIS, CD68+ and CD115+ expression was shown to be associated with ER- negativity, HER2-positivity, and high grade [46,49]. Chen et al. showed that the high frequency of CD163+ macrophages was significantly associated with higher risk of DCIS recurrence [46]. Interestingly, increased risk of DCIS recurrence was also found to correlate with a high density of CD115+ cells accompanied by low and high numbers of CD8+HLADR+ cells (activated CD8+ cells) and CD8+HLADR- cells (non-activated CD8+ cells), respectively [49]. This suggests that CD115+ cells may affect the activation state of CD8+ cells and consequently alter their prognostic significance. Available evidence relating macrophage infiltration to microinvasion is currently both inconsistent and inconclusive [33,46,49]. A possible correlation between the molecular context and macrophages’ influence on DCIS progression was recently shown in a HER2-positive DCIS model in mouse (MMTV-HER2). The authors demonstrated that when macrophages were depleted during asymptomatic pre-malignant stages, the time to tumour detection was delayed. This implies that in early stages of HER2-positive lesions, the macrophages might promote tumour progression [85].

## 5. Crosstalk between HER2 Pathway and Immune Signalling

Recent studies in several BC biological settings offer some mechanistic explanations for the link between the density of immune infiltrates, HER2-positive molecular subtype, and clinical course.

Triuzli et al., using gene expression and immunohistochemistry, stratified HER-positive IDCs into trastuzumab-resistant and trastuzumab-sensitive tumours, and showed that the latter had significantly higher levels of CC and CXCL chemokines (involved in the recruitment of monocytes and migration of T and B cells, respectively), a denser infiltration of CD8+ and CD68+ cells, and higher levels of PD-L1 and PD-L2 [86]. Using an in vitro model, the authors demonstrated that CCL2, mainly involved in the recruitment of monocytes, was modulated by the PI3K/AKT/NF-kB pathway downstream of HER2. Inhibition of NF-kB activity blocked the HER2-mediated production of CCL2, which, in an in vivo model, was further found to impair recruitment of immune cells and reduce tumour sensitivity to trastuzumab [86]. Moreover, inhibition of the HER2 signal significantly reduced, at both the mRNA and protein level, the expression of PD-L1 and PD-L2 in tumour cells, thus implicating HER2 in the modulation of innate immune resistance [86].

A central role of the HER2/PI3K/AKT/NF-kB signalling axis in immune responses to the HER2 oncodrive was confirmed by another study where activation of the HER2→ NF-κB signalling in the inflammatory milieu of IDC initiated and sustained the inflammation that supported the generation and maintenance of breast cancer stem cells (BCSC) [51]. Using an in vitro DCIS model, we recently demonstrated that inflammatory stimuli activated the HER2→NF-κB/COX2/HIF1-α pathway, found to confer a growth advantage to HER2-negative cells. Given the heterogeneity of DCIS lesions and the predominance of both HER2-negative IDC and invasive recurrences, this mechanism is likely to contribute to disease progression or relapse [23].

Recurrence of tumour typically occurs after a certain time after removal of the primary tumour. Neither the cells of origin nor the stimuli leading to tumour regrowth have been identified but the phenomenon most likely involves a state of dormancy in the subset of tumour cells, an expansion of cancer stem cells, and/or the presence of circulating tumour cells [87]. The role of the immune system in these cellular events is unknown but it can be speculated that for HER2-positive cells, in particular, an inflammatory environment promotes NF-κB-mediated signalling that drives tumour regrowth. A schematic representation of such a permissive environment where the HER2→NF-κB axis would be fundamental for tumour outcome is depicted in Figure 1.

Several studies demonstrated an impact of a crosstalk between pathways mediated by HER2 and certain chemokines on various aspects of BC cell behaviour. For example, using a two-chamber co-culture system, Kang et al. showed that CCL2 secreted from macrophages overexpressing metalloproteinase 11 (MMP11) activated the MAPK pathway via its receptor CCR2 in HER2+ BC cells, which, in turn, upregulated MMP9 and led to BC cell migration and recruitment of monocytes [88]. Castiello et al. demonstrated that the interaction between HER2 and IFN-I controlled the stemness of BC cells. Disruption of IFN-1 signalling (via non-functional mutation in the IFN1I receptor) in tumours arising in HER2/neu transgenic mice deregulated a number of genes, including aldehyde dehydrogenase-1A1 (ALDH1A1), the marker of breast cancer stem cells. In vitro exposure of wild-type (neuT+) mammospheres and cell lines to antibodies against IFN-I resulted in an increased frequency of ALDH+ cells [89].

In summary, increasing evidence of a crosstalk between the HER2 and immune pathways in an early stage of BC development suggests that the search, which only just started, for its key regulators and underlying mechanisms may bring results of major clinical implications.

## 6. Conclusions

Available evidence indicates that the tumour immune microenvironment plays an important role in breast cancer evolution. In both DCIS and IDC, different frequencies of immune cells are found in different BC subtypes, indicating that their impact on disease outcome is determined by the subtype-specific interaction with tumour cells. In particular, in HER2-positive DCIS, the mechanism underlying this crosstalk might be of paramount significance for the estimation of the anticipated biological behaviour of the tumour and, hence, evaluation of the risk for disease progression/recurrence. This further implies that assessment of the immune signature of the stroma, along with HER2 expression in DCIS, might provide a better guideline for treatment options, thus making a compelling case for re-assessment of the routine procedure of prognostication in DCIS patients.

## Figures and Tables

**Figure 1 biomedicines-10-01061-f001:**
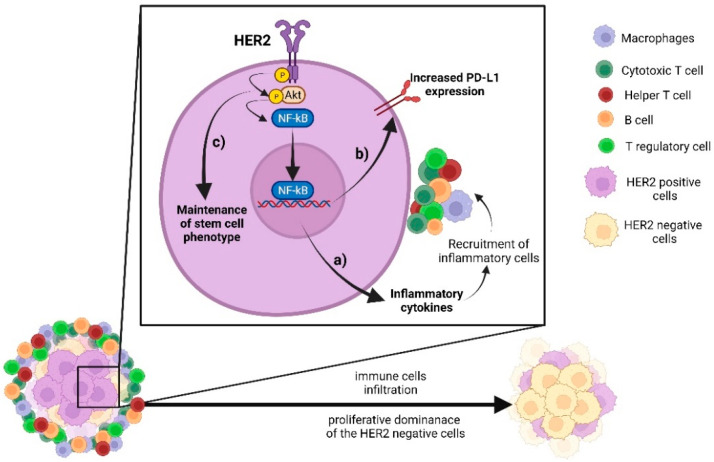
Crosstalk between HER2 pathway and immune signalling. HER2 signalling activates the AKT→ NF-κB axis, which stimulates: (**a**) production of inflammatory cytokines and infiltration of immune cells; (**b**) increase of PD-L1 expression; and (**c**) maintenance of stem cell phenotype. All those phenomena may also confer proliferative dominance of HER2-negative cells, leading to HER2-negative invasive recurrence/progression.

## Data Availability

Not applicable.

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
