# Peer review of "Subtype-Specific Tumour Immune Microenvironment in Risk of Recurrence of Ductal Carcinoma In Situ: Prognostic Value of HER2"

_biomedicines, 2022, doi:10.3390/biomedicines10051061_

Round 1

Reviewer 1 Report

This is a very well written review addressing multiple factors involved in the immune microenvironment of DCIS tumors that contribute to risk of recurrence. I have only very minor edits to suggest.

  1. Section 2, like 82 - in Triple negative, suggest either making the t small or the N capital to keep consistent with the other subcategories.
  2. Table 1 - adjust column spacing so words are not divided in 2 lines
  3. Section 4 - CD8+ and CD4+ has inconsistent spacing following each -- some have spaces between the + and others do not. Please make consistent.
  4. Section 4.3 header - remove the space between (T
  5. Section 4 last paragraph first word - please change to several, not sever

Author Response

Reviewer 1

  1. Section 2, like 82 - in Triple negative, suggest either making the t small or the N capital to keep consistent with the other subcategories.
  2. Table 1 - adjust column spacing so words are not divided in 2 lines
  3. Section 4 - CD8+ and CD4+ has inconsistent spacing following each -- some have spaces between the + and others do not. Please make consistent.
  4. Section 4.3 header - remove the space between (T
  5. Section 4 last paragraph first word - please change to several, not sever

Authors:

All inaccuracies specified in 1-4. have been corrected.

Reviewer 2 Report

In their review, Solek and colleagues illustrate the potential predictive role of the tumor immune microenvironment in defining patient outcome and tumor aggressiveness. The authors included the immune characterization of different molecular subtypes of breast carcinoma, followed by the literature about the single immune cell populations (e.g. CD8+ T Lymphocytes, CD4+, B-cells, etc). The topic is timely and relevant, however, this review has several main issues.

The authors chose to illustrate too many topics, without going into details. Each argument is just “listed” as subsequent lines of different findings from different groups on different tumor subtypes. The reader is quite confused because the final message is not clear.

So, I suggest to the authors revise the entire setting of the work, by choosing one argument and going more into detail. For example, the authors can decide to explore the different subtypes of breast cancer and remove the part of HER2 which is quite meaningless as it is in the present form. Or, they can focus on HER2 and go deeper.

Here are many suggestions to improve the manuscript:

  • There are many errors in the immune cell classification. The authors must be more rigorous.
  • For example, in lines 52-53 the authors claimed that CD4+ cells correlated with the number of TILs. TILs include both CD4, CD8 and B-cells (which one the authors are referring to?).
  • In line 128, the authors wrongly claimed that myeloid-derived suppressor cells are macrophages. This is a misleading concept and should be carefully rephrased.
  • In line 133, “TAMs can be polarized”, again this is a misleading concept. Macrophages can polarize in M1/M2 subtypes, but in this sentence, it seems that this is a particular feature of TAMs. Moreover, the authors should add the different immunophenotyping information about these subtypes.
  • TAMs are typically identified by CD68, CD163, CD115 [83]. I checked the reference and this is not mentioned. CD68 is expressed by all macrophages. CD163 and CD115 are more related to TAMs with M2 polarization.
  • Paragraph 2 (molecular subtypes of breast carcinoma) should be improved. All different molecular subtypes should be described. For example, Triple-negative or basal-like subtypes are not mentioned in the description of the characteristics of the tumor microenvironment. The authors can also add a schematic figure to illustrate the difference between subtypes.
  • Paragraph 4.1 (CD8 T cell). It is just a list of different findings. The authors should try to improve the flow and focalize on the message. Maybe it is better to illustrate the different results by grouping pro-tumoral vs anti-tumoral effects of different CD8 subpopulations.
  • Memory CD8 T cells should be included in the text since their presence and localization seem important in breast tumor immunity and in determining patient outcomes (doi: 1172/jci.insight.130000)
  • Paragraph 4.4. Markers of immune exhaustion of T cells. The authors mentioned only PD1 and some immune-checkpoint molecules. However, a broad literature suggests a more complex profile of exhaustion due to recent single-cell RNA-sequencing results (for example https://doi.org/10.1038/s41467-021-23324-4, DOI: 1007/978-1-0716-1507-2_6).
  • Paragraph 4.5. Tumor-infiltrating B cells. This paragraph is very skinny. B-cells have a very complex role in mediating both pro-tumor and anti-tumor activities due to the different contexts. This deserves to be further explained. For example, it is well reported a prognostic role in TNBC patients or HER2+ patients, which is associated with a good outcome (e.g. DOI: 10.1007/978-1-0716-1507-2_6, doi: 1172/jci.insight.129641)

Minor points:

  • Line 79-85 should be moved to the Introduction
  • Many references are wrongly cited (e.g. ref 42, 29 – the first author does not correspond to the one cited)
  • Please add the reference numbers in the table instated of just the name and the year of publication
  • English should be revised by an English native speaker

Author Response

Comments and Suggestions for Authors

In their review, Solek and colleagues illustrate the potential predictive role of the tumor immune microenvironment in defining patient outcome and tumor aggressiveness. The authors included the immune characterization of different molecular subtypes of breast carcinoma, followed by the literature about the single immune cell populations (e.g. CD8+ T Lymphocytes, CD4+, B-cells, etc). The topic is timely and relevant, however, this review has several main issues.

The authors chose to illustrate too many topics, without going into details. Each argument is just “listed” as subsequent lines of different findings from different groups on different tumor subtypes. The reader is quite confused because the final message is not clear.

So, I suggest to the authors revise the entire setting of the work, by choosing one argument and going more into detail. For example, the authors can decide to explore the different subtypes of breast cancer and remove the part of HER2 which is quite meaningless as it is in the present form. Or, they can focus on HER2 and go deeper.

Authors:

As specified in the title, HER2 and its prognostic role in DCIS recurrence is the main focus of the review. We felt that, in the context of the recent clinical trial relating overexpression of HER2 with risk of recurrence (Thorat, M.A. et al.,   2021), summarising available evidence of    the prognostic value of TIME in HER2-positive subtype in relation to  available mechanistic data might add  a new dimension to its significance in DCIS.

We altered the closing fragment of the Introduction to make this goal more transparent. It reads now:

Here, an updated review of existing data on a role of the HER2-TIME crosstalk in DCIS and its potential value in stratification of risk of recurrence is presented in the context of available evidence of  subtype-dependent clinical significance of stromal immune cells in DCIS (summarized in Table 1). – line 83-86

Here are many suggestions to improve the manuscript:

  1. There are many errors in the immune cell classification. The authors must be more rigorous.
  1. For example, in lines 52-53 the authors claimed that CD4+ cells correlated with the number of TILs. TILs include both CD4, CD8 and B-cells (which one the authors are referring to?).

Authors:

This statement and citation were incorrect and have now been removed.

  1. In line 128, the authors wrongly claimed that myeloid-derived suppressor cells are macrophages. This is a misleading concept and should be carefully rephrased.

Authors:

We corrected this fragment and it reads now:

Tumour-associated macrophages (TAMs),  derived from mononuclear cells, are the most abundant population of tumour-infiltrating immune cells in TME [80]. - line 284-285

  1. In line 133, “TAMs can be polarized”, again this is a misleading concept. Macrophages can polarize in M1/M2 subtypes, but in this sentence, it seems that this is a particular feature of TAMs. Moreover, the authors should add the different immunophenotyping information about these subtypes.

Authors:

This has been corrected as follows:

Macrophages display high phenotypic and functional plasticity. They have two polarization states: classically activated M1 and alternatively activated M2 subtypes, that for TAMs, have been associated with anti- and pro-tumorigenic activities, respectively. - line 289-292

  1. TAMs are typically identified by CD68, CD163, CD115 [83]. I checked the reference and this is not mentioned. CD68 is expressed by all macrophages. CD163 and CD115 are more related to TAMs with M2 polarization.

Authors:

The reference has been replaced by “Bronte, V., Brandau, S., Chen, SH. et al. Recommendations for myeloid-derived suppressor cell nomenclature and characterization standards. Nat Commun 7, 12150 (2016). [83]

We apologize for this mistake.

  1. Paragraph 2 (molecular subtypes of breast carcinoma) should be improved. All different molecular subtypes should be described. For example, Triple-negative or basal-like subtypes are not mentioned in the description of the characteristics of the tumor microenvironment. The authors can also add a schematic figure to illustrate the difference between subtypes.

Authors:

As we feel that BC subtyping is common knowledge, to avoid being condescending  we expanded the paragraph by only several key features relating to the subtypes and an additional reference (Https://pubmed.ncbi.nlm.nih.gov/33011829/).

It reads now:

Based on a seminal study by Perou et al., four main molecular subtypes of BC, related to expression of estrogen receptor (ER),  progesterone receptor (PR) and HER2 (ER/PR/HER2 status), have been distinguished: Luminal A, Luminal B, HER2-positive and Triple Negative (TN) [42]. All these subtypes are present in both DCIS and IDC albeit with different frequencies. In both DCIS and IDC, HER2-positive and TN subtypes are frequently associated with high histopathological grade and several genetic events such as p53 dysfunction, aneuploidy, Ras protein overexpression, and poor prognosis while luminal subtypes are usually found to be of low grade [24,42]. - line 89-97

  1. Paragraph 4.1 (CD8 T cell). It is just a list of different findings. The authors should try to improve the flow and focalize on the message. Maybe it is better to illustrate the different results by grouping pro-tumoral vs anti-tumoral effects of different CD8 subpopulations.

Authors:

As suggested, for better clarity, the paragraph has been rearranged.

  1. Memory CD8 T cells should be included in the text since their presence and localization seem important in breast tumour immunity and in determining patient outcomes (doi: 1172/jci.insight.130000)

Authors:

According to the recent reports, memory CD8T cells indeed emerge as critical determinants of outcome of patients with IDC (specifically TNBC).  However, as evidence recognising their role  in DCIS is currently lacking, a short description of these cells (supported by the reference suggested by the Reviewer) has been added at the end of the paragraph to illustrate the complexity of the population.

It reads now:

However, it should be emphasized that CD8+ TILs represent a heterogenous population and little is known about clinical significance of their functional status or spatial location. For example, CD8+ TILs positive for CD103, so called tissue-resident memory T cells (TRMs) have been recently identified as key immune players in TME. Their increased densities in cancer island within IDC were significantly association with relapse-free survival (RFS), but their prognostic value in DCIS is still to be revealed [62]. – line 189-195

  1. Paragraph 4.4. Markers of immune exhaustion of T cells. The authors mentioned only PD1 and some immune-checkpoint molecules. However, a broad literature suggests a more complex profile of exhaustion due to recent single-cell RNA-sequencing results (for example https://doi.org/10.1038/s41467-021-23324-4 DOI: 1007/978-1-0716-1507-2_6).

Authors:

We agree with this comment and, although available findings do not relate to the prognostication in DCIS, we have included this important study to illustrate degree of diversity of immune cells and their states, undoubtedly relevant also to DCIS biology.

It reads now: 

This is most likely due to the heterogeneity of TIL population in terms of distinct states and profiles of exhaustion, definition of which remains a major challenge. Results of recent studies using single-cell RNA sequencing confirm a remarkable diversity of immune cells suggesting that distinct TIL states may have different impact on disease course and outcome [78].  line 262-267

  1. Paragraph 4.5. Tumor-infiltrating B cells. This paragraph is very skinny. B-cells have a very complex role in mediating both pro-tumor and anti-tumor activities due to the different contexts. This deserves to be further explained. For example, it is well reported a prognostic role in TNBC patients or HER2+ patients, which is associated with a good outcome (e.g. DOI: 10.1007/978-1-0716-1507-2_6, doi: 1172/jci.insight.129641)

Authors:

Similarly to 8) and 9), while the role of  TIL-B cells in immune responses and their impact on disease outcome are well characterized in IDC (e.g. references suggested by the Reviewer), existing data on  their activity and significance in DCIS are scarce. In addition, as available reports show, in contrast to IDC, TIL-B cells DCIS are associated with poor prognosis.

This paragraph has been expanded to highlight this point and it reads now: 

In contrast to DCIS, higher densities of TILBs detected in IDC have been associated with better survival, especially in ER-negative, TNBC and HER2-positive tumours.– line 279-281

Minor points:

  • Line 79-85 should be moved to the Introduction

Authors:

These are the Introduction.

  • Many references are wrongly cited (e.g. ref 42, 29 – the first author does not correspond to the one cited)

Authors:

This has been corrected.

  • Please add the reference numbers in the table instated of just the name and the year of publication

Authors:

The table has been corrected accordingly.

  • English should be revised by an English native speaker

Authors:

This has been done.

References added:

  1. Sadeghalvad M, Mohammadi-Motlagh HR, Rezaei N. Immune microenvironment in different molecular subtypes of ductal breast carcinoma. Breast Cancer Res Treat. 2020.
  2. Egelston CA, Avalos C, Tu TY, Rosario A, Wang R, Solomon S, et al. Resident memory CD8+ T cells within cancer islands mediate survival in breast cancer patients. JCI Insight. 2019;4(19).

  1. Andreatta M, Corria-Osorio J, Muller S, Cubas R, Coukos G, Carmona SJ. Interpretation of T cell states from single-cell transcriptomics data using reference atlases. Nat Commun. 2021;12(1):2965.
  2. Bronte, V.; Brandau, S.; Chen, S.H.; Colombo, M.P.; Frey, A.B.; Greten, T.F.; Mandruzzato, S.; Murray, P.J.; Ochoa, A.; Ostrand-Rosenberg, S.; et al. Recommendations for Myeloid-Derived Suppressor Cell Nomenclature and Characterization Standards. Nat. Commun. 2016, 7, doi:10.1038/ncomms12150.

Round 2

Reviewer 2 Report

Accept in the present form.